# Machine Learning Models Integrating Dietary Indicators Improve the Prediction of Progression from Prediabetes to Type 2 Diabetes Mellitus

**DOI:** 10.3390/nu17060947

**Published:** 2025-03-08

**Authors:** Zhuoyang Li, Yuqian Li, Zhenxing Mao, Chongjian Wang, Jian Hou, Jiaoyan Zhao, Jianwei Wang, Yuan Tian, Linlin Li

**Affiliations:** 1Department of Epidemiology and Health Statistics, College of Public Health, Zhengzhou University, Zhengzhou 450001, China; tz19990102@163.com (Z.L.); maozhr@gmail.com (Z.M.); tjwcj2005@126.com (C.W.); 13667176505@163.com (J.H.); 17839948980@163.com (J.Z.); 15716690965@163.com (J.W.); ty163email@163.com (Y.T.); 2Department of Clinical Pharmacology, School of Pharmaceutical Science, Zhengzhou University, Zhengzhou 450001, China; liyuqian0214@126.com

**Keywords:** type 2 diabetes mellitus, prediabetes, diet, machine learning, prediction model

## Abstract

**Background**: Diet plays an important role in preventing and managing the progression from prediabetes to type 2 diabetes mellitus (T2DM). This study aims to develop prediction models incorporating specific dietary indicators and explore the performance in T2DM patients and non-T2DM patients. **Methods**: This retrospective study was conducted on 2215 patients from the Henan Rural Cohort. The key variables were selected using univariate analysis and the least absolute shrinkage and selection operator (LASSO). Multiple predictive models were constructed separately based on dietary and clinical factors. The performance of different models was compared and the impact of integrating dietary factors on prediction accuracy was evaluated. Receiver operating characteristic (ROC) curve, calibration curve, and decision curve analysis (DCA) were used to evaluate the predictive performance. Meanwhile, group and spatial validation sets were used to further assess the models. SHapley Additive exPlanations (SHAP) analysis was applied to identify key factors influencing the progression of T2DM. **Results**: Nine dietary indicators were quantitatively collected through standardized questionnaires to construct dietary models. The extreme gradient boosting (XGBoost) model outperformed the other three models in T2DM prediction. The area under the curve (AUC) and F1 score of the dietary model in the validation cohort were 0.929 [95% confidence interval (CI) 0.916–0.942] and 0.865 (95%CI 0.845–0.884), respectively. Both were higher than the traditional model (AUC and F1 score were 0.854 and 0.779, respectively, *p* < 0.001). SHAP analysis showed that fasting plasma glucose, eggs, whole grains, income level, red meat, nuts, high-density lipoprotein cholesterol, and age were key predictors of the progression. Additionally, the calibration curves displayed a favorable agreement between the dietary model and actual observations. DCA revealed that employing the XGBoost model to predict the risk of T2DM occurrence would be advantageous if the threshold were beyond 9%. **Conclusions**: The XGBoost model constructed by dietary indicators has shown good performance in predicting T2DM. Emphasizing the role of diet is crucial in personalized patient care and management.

## 1. Introduction

Diabetes is a chronic metabolic disorder with a rapidly increasing global prevalence. In 2019, 463 million people worldwide were living with diabetes [1]. By 2021, this number had risen to 536.6 million and is projected to reach 700 million by 2045 [2]. Diabetes is commonly classified into different types based on its etiology and clinical characteristics, including type 1 diabetes mellitus, type 2 diabetes mellitus (T2DM), gestational diabetes mellitus, and specific types of diabetes due to other causes [3]. T2DM is the predominant form of diabetes, accounting for more than 90% of all cases [4]. T2DM primarily affects adults, especially middle-aged and elderly individuals, and is characterized by a strong genetic predisposition. It is also significantly influenced by environmental factors, including poor dietary habits, physical inactivity, and obesity. The pathogenesis of T2DM mainly involves insulin resistance and insufficient insulin secretion [4]. Multiple studies have shown that individuals with prediabetes are at a higher risk of developing type 2 diabetes mellitus (T2DM) compared to those with normal blood glucose levels [5,6]. In 2021, 197 million adults in China were diagnosed with prediabetes [2]. The conversion rate among Chinese adults ranges from 7% to 15% [7,8,9,10]. Therefore, early intervention and management should be implemented in patients with prediabetes to effectively prevent the onset of T2DM.

The progression from prediabetes to T2DM is a complex process influenced by various genetic, lifestyle, and metabolic factors. Previous studies have found that certain specific genes, such as antibiotic resistance genes, may increase an individual’s risk of T2DM by influencing insulin resistance [11]. Lifestyle factors, including physical inactivity, overweight, and obesity can further exacerbate insulin resistance [12,13,14]. Additionally, factors such as age, gender, anxiety, depression, and cardiovascular diseases are key contributors to this process [15,16,17,18]. Furthermore, a growing body of research highlights the significant impact of dietary factors on the development of T2DM. Diet not only influences weight management and insulin sensitivity but also directly regulates blood glucose levels [19]. Diets high in refined carbohydrates and saturated fats increase the risk of T2DM, while diets rich in fiber, healthy fats, and moderate protein help reduce this risk [20,21,22]. Therefore, exploring the relationship between dietary factors and the progression of prediabetes is crucial for developing effective intervention strategies.

Given the significant role of diet in the progression, integrating dietary factors into predictive models may improve the assessment of T2DM risk in individuals with prediabetes. Most current predictive models primarily consider traditional risk factors such as body mass index (BMI), blood glucose levels, and insulin resistance, neglecting the influence of dietary factors [23]. Dietary indicators not only reflect long-term metabolic status but also serve as modifiable lifestyle factors with significant clinical value. Although Nicolaisen et al. incorporated self-reported dietary data into their predictive model, they did not systematically evaluate the effects of specific dietary components [24].

Advancements in data science have made machine learning (ML) a powerful tool for enhancing the performance of predictive models. In recent years, the application of ML in the medical field has grown, especially in disease prediction and risk assessment. For example, Yang et al. developed a logistic regression model to predict diabetes using hospital physical examination data, achieving an AUC of 0.787 [25]. Yokota et al. similarly developed a multivariable logistic regression model to predict the progression of T2DM, with an AUC of 0.80, highlighting its reliability in forecasting diabetes progression [26]. Although logistic regression models demonstrate predictive capabilities in some studies, traditional models often fail to capture the complex nonlinear relationships in a multifactorial condition. In contrast, models such as random forest, support vector machine, and extreme gradient boosting are more effective at capturing nonlinear relationships and interactions between variables. For example, Qing et al. applied four ML models (logistic regression, decision tree, random forest, and extreme gradient boosting) in an elderly population, showing that extreme gradient boosting outperformed the others, with an AUC of 0.6707 [10]. Additionally, among four ML models based on physical examination indicators (e.g., blood pressure, weight, and BMI), Chen et al. found that logistic regression performed best, with an AUC of 0.74 [27]. Although logistic regression performed well in this study, other complex algorithms remain promising. By comparing different ML models, the most suitable model can be selected to improve prediction accuracy. Moreover, the application of ML in early diabetes diagnosis has notably improved prediction accuracy, especially when dealing with multiple clinical indicators and complex data. Zou et al. combined several important clinical indicators (e.g., fasting blood glucose, glycosylated haemoglobinA1c, high-density lipoprotein cholesterol, and triglycerides) and used extreme gradient boosting for prediction [28]. The integration of multiple data sources has become a prominent research trend in recent years. Jiang et al. used support vector machine, random forest, and extreme gradient boosting models combined with nuclear magnetic resonance metabolomics data to identify nine key metabolites. Their method achieved an AUC of 0.823, significantly surpassing traditional approaches that relied solely on clinical data [29]. Despite considerable progress in improving diabetes prediction capabilities, dietary factors remain underexplored. Based on the background above, this study aims to explore the impact of dietary factors on the progression of prediabetes to T2DM and to develop predictive models using ML techniques. By integrating dietary factors with traditional clinical data (e.g., body mass index and fasting plasma glucose), we aim to enhance the accuracy of assessing the risk of T2DM in individuals with prediabetes. Additionally, this study will compare different ML models to identify the optimal model for enhancing prediction accuracy and supporting early intervention strategies.

## 2. Materials and Methods

### 2.1. Study Design

The dataset for this study was derived from the Henan Rural Cohort Study in China. The cohort was established across five rural regions of Henan province (south, central, north, east, and west) [30]. The baseline survey was conducted from July 2015 to September 2017, and the first follow-up was completed by the end of September 2022. Baseline data were collected from 39,259 participants using standardized questionnaires that covered demographics, lifestyle, sleep patterns, personal and family medical history, menstrual and reproductive history (for females), and mental health status. Trained doctors, nurses, and technicians collected blood, urine, and selected fecal samples, as well as performed anthropometric measurements and clinical examinations.

This study enrolled 2669 prediabetic individuals and selected 32 relevant variables based on a literature review. These variables included general demographic characteristics (age, gender, education level, marital status, and income level); lifestyle factors (dietary intake, smoking, alcohol consumption, and physical activity); sleep quality; personal medical history (hypertension, cancer, and kidney failure); family medical history; biochemical parameters (fasting plasma glucose, total cholesterol, triglycerides, high-density lipoprotein cholesterol, and low-density lipoprotein cholesterol); and anthropometric measurements (systolic blood pressure, diastolic blood pressure, heart rate, body mass index, and waist circumference). After excluding 358 lost to follow-up, 90 deaths, and six participants with missing dietary data, 2215 participants were included in the final analysis.

### 2.2. Definitions of Prediabetes and T2DM

According to the American Diabetes Association (ADA) diagnostic criteria (2009) [4], T2DM was defined by a fasting plasma glucose (FPG) level of ≥7.0 mmol/L or an HbA1c level of ≥6.5%. In addition, a self-reported history of T2DM with the present use of antidiabetic drugs after excluding all other types of diabetes was also included in the diagnostic criteria for T2DM in this study. Similarly, individuals with 6.1 ≤ FPG < 7.0 mmol/L or 5.7 ≤ HbA1b < 6.5% were ascertained as prediabetes after excluding other disorders with impaired plasma glucose regulation.

### 2.3. Data Preprocessing

Data preprocessing involves cleaning, transforming, and standardizing data to improve its suitability for machine learning algorithms or statistical analyses. This process enhances model performance and increases the credibility of the results. In this study, no variable had more than 25% missing data (Appendix A). Missing data were addressed using multiple imputation based on the random forest method. Data normalization was performed using “StandardScaler” from “sklearn.preprocessing”. Variables showing statistical significance in the univariate analysis were incorporated into least absolute shrinkage and selection operator (LASSO) regression to select the final model variables. LASSO selects variables by shrinking unimportant predictor coefficients to zero through penalization. The penalty strength is controlled by the parameter λ and the optimal λ is determined through 10-fold cross-validation. The dataset contains 1841 instances in the negative class and 374 in the positive class, with a ratio of approximately 4.92, indicating class imbalance. This imbalance in the outcome variable may lead the model to misclassify most cases, negatively impacting prediction accuracy. Therefore, the Synthetic Minority Over-sampling Technique (SMOTE) was applied to address the data imbalance [31].

### 2.4. Model Training

The 2215 participants in this study were split into training and testing sets with a 6:4 ratio. Four classification models were evaluated to identify the most suitable one for this study. The models compared were logistic regression (LR), support vector machine (SVM), random forest (RF), and extreme gradient boosting (XGBoost). LR is a probability-based linear classification algorithm that uses the sigmoid function to transform linear predictions into class probabilities. SVM is a classification algorithm that maximizes geometric margins to find the optimal hyperplane for linearly or nonlinearly separating data. RF is an ensemble-based decision tree algorithm that constructs multiple decision trees and aggregates their predictions for classification or regression. XGBoost is an ensemble learning algorithm based on gradient boosting that iteratively trains decision trees and optimizes the loss function using gradient descent to enhance model performance. Grid search and cross-validation were used to find the optimal combination of hyperparameters. The training and validation process was repeated ten times to ensure that data from each group were validated. All statistical analyses and machine learning models were performed using Python 3.11 and R 4.4.1.

### 2.5. Model Evaluation

The predictive performance of the machine learning algorithms was evaluated on an independent test set. The confusion matrix was used to calculate metrics for classification performance, including accuracy, specificity, precision, recall, and F1 score. Model performance was visualized using the Receiver Operating Characteristic (ROC) curve [32], calibration curve, and decision curve analysis (DCA). The calibration curve is a plot based on the predicted probabilities of the model and the actual outcomes. Ideally, the calibration curve should be close to the 45-degree diagonal line. DCA is used to evaluate the clinical utility of predictive models. It calculates net benefit by combining model predictions with threshold probabilities. If the model’s net benefit is higher than “treat all” or “treat none” within the threshold range, the model is considered clinically useful.

### 2.6. Model Interpretation

Shapley Additive Explanations (SHAP) was used to understand the influence of features on model outputs and identify the most significant features [33]. SHAP is based on the concept of Shapley values from game theory, which quantifies the contribution of each feature to the model’s decision-making process. Specifically, SHAP computes the marginal contribution of each feature to the model’s output under different conditions, providing an intuitive importance score for each prediction. These scores identify the features with the greatest impact on the decision-making process.

## 3. Results

A total of 2215 prediabetic patients were included in the final analysis (Figure 1). During an average follow-up of 3.84 years, 374 participants (16.9%) developed T2DM. Table 1 shows the baseline characteristics of all participants. Participants who developed T2DM were fewer across different educational levels and drinking status compared to the other groups (both *p* < 0.05). They had lower total cholesterol (TC) and high-density lipoprotein cholesterol (HDL-C), but higher FPG and triglycerides (TG) (all *p* < 0.05). Additionally, participants who developed T2DM had lower heart rate than those who returned to normal glucose tolerance (NGT) (*p* < 0.05). Appendix A presents a comparison of the data before and after imputation. No statistically significant differences were observed in any variables before and after imputation (*p* > 0.05), indicating that the imputation was effective.

### 3.1. T2DM Prediction Model Based on Traditional Indicators

Univariate analysis identified eight indicators, including age, income level, FPG, TC, TG, HDL-C, BMI, and waist circumference (WC), having a significant impact on the outcome (*p* < 0.05) (Appendix A). Figure 2 displays the results of the LASSO regression. LASSO regression reduced the number of variables predicting the outcome to seven. The selected variables were age, income level, FPG, TC, TG, HDL-C, and BMI. The optimal lambda.min determined through cross-validation was 0.001285927. Appendix A summarizes the seven non-zero coefficients identified by LASSO at the lambda.min value. In this study, all variables had variance inflation factor (VIF) values below 10, indicating no multicollinearity, as shown in Appendix A.

The seven selected indicators were used to construct four different models for T2DM prediction. Figure 3A and Appendix A show the ROC curves and performance metrics of the predictive models. The XGBoost model outperformed the others in area under the curve (AUC), specificity, and precision, achieving an AUC of 0.854 (95% CI: 0.835–0.872), specificity of 0.772 (95% CI: 0.742–0.803), and precision of 0.778 (95% CI: 0.749–0.806). The RF model demonstrated higher accuracy, recall, and F1 score, with an accuracy of 0.779 (95% CI: 0.757–0.798), recall of 0.834 (95% CI: 0.806–0.860), and F1 score of 0.793 (95% CI: 0.772–0.811). The two models performed similarly across various metrics, indicating that both XGBoost and RF demonstrated strong predictive performance for T2DM based on traditional indicators. Calibration curves indicated a high degree of consistency between the predicted and actual probabilities of the two models (Appendix A).

### 3.2. T2DM Prediction Model Based on Traditional Indicators and Dietary Indicators

Appendix A lists the newly included dietary indicators: refined grains, red meat, white meat, eggs, fruits, vegetables, beans, nuts, and whole grains. The three groups, classified by glycemic status at follow-up, showed significant differences in refined grains, eggs, vegetables, and whole grains intake. The prediabetic group consumed fewer whole grains than the other two groups (all *p* < 0.05). The nine dietary indicators were combined with seven traditional indicators (age, income level, FPG, TC, TG, HDL-C, and BMI) to construct a new predictive model. Figure 3B and Appendix A show the ROC curves and performance metrics of the predictive models. The XGBoost model outperformed others, with the highest AUC of 0.929 (95% CI: 0.916–0.942), accuracy of 0.868 (95% CI: 0.849–0.886), specificity of 0.900 (95% CI: 0.879–0.920), precision of 0.895 (95% CI: 0.873–0.916), and F1 score of 0.865 (95% CI: 0.845–0.884). The SVM model had the highest recall at 0.839 (95% CI: 0.813–0.864). The DeLong test showed a significant AUC improvement in the XGBoost model after adding dietary indicators (*p* < 0.001) (Appendix A). The calibration curves of the four models were shown in Appendix A. The predicted probability curves of the XGBoost and SVM models closely matched the optimal curve.

As shown in Appendix A, the XGBoost model demonstrated the best clinical utility across the threshold range from 9% to 100%. The RF model also provided a high net benefit within the range of 12% to 96%. In contrast, the LR model showed the lowest clinical utility, with notable effectiveness only within the threshold range of 19% to 71%. When the threshold was between 78% and 90%, the LR model’s net benefit was lower than that of the “treat none” strategy.

Given the XGBoost-based T2DM prediction model’s excellent performance in prediction accuracy, DCA, and calibration curves, we used SHAP to interpret its results. Figure 4A shows that FPG has the greatest impact on the progression from prediabetes to T2DM. Apart from FPG, the other variables influencing T2DM progression were eggs, whole grains, income level, red meat, nuts, HDL-C, and age. In Figure 4B, the relative contributions of each feature to the XGBoost model showed a positive or negative correlation with T2DM progression. FPG had a positive impact on the progression to T2DM, while other features, such as whole grains, income level, and HDL-C, had a negative impact.

To understand how features influence each patient’s risk prediction, we conducted a personalized risk analysis using SHAP for two patients: one who did not progress to T2DM and one who did (Figure 5). The baseline value represented the average prediction from the XGBoost model based on the training data. The end of the chart showed the model’s final prediction for the specific sample. Figure 5A shows the personalized risk factor analysis for a prediabetic patient who progressed to T2DM. The baseline value was −0.02. Consumption of eggs (110 g), refined grains (611 g), age (57 years), and red meat (13 g) increased the prediction by 1.11, 0.54, 0.42, and 0.4, respectively. Conversely, TG (0.849 mmol/L) decreased the prediction by 0.49, resulting in a final prediction value of 2.711. Figure 5B shows the personalized risk factor analysis for a prediabetic patient who did not progress to T2DM. Consumption of eggs (0 g), FPG (6.23 mmol/L), TG (0.96 mmol/L), and nuts (0 g) decreased the prediction by 0.62, 0.49, 0.42, and 0.40, respectively. Conversely, red meat (5 g) increased the prediction by 0.39, resulting in a final prediction value of −1.036.

### 3.3. Sensitivity Analysis

To confirm the reliability of selecting XGBoost as the most suitable model, we divided the data into subgroups based on gender and region for both group and spatial validation. The regional data were split into east, south, west, and north of Henan province. Appendix A show the prediction performance for each subgroup. The results indicated that the XGBoost model showed the highest robustness across most subgroups.

## 4. Discussion

Prediabetes is a reversible condition associated with a high risk of developing T2DM. Early prediction of the progression from prediabetes to T2DM is essential for effective prevention and intervention strategies. Although diet is a critical factor influencing diabetes risk, its integration into predictive models has been insufficiently studied and applied. To our knowledge, this study is the first to incorporate specific dietary indicators into ML models designed to predict the progression from prediabetes to T2DM. Our study found that the XGBoost model consistently demonstrated excellent predictive performance in the test set, cohort validation, and spatial validation, both before and after adding dietary indicators. Furthermore, SHAP analysis further revealed that factors like FPG, egg consumption, whole grains, income level, red meat, nuts, HDL-C, and age significantly contributed to the XGBoost model’s predictions.

Individuals with prediabetes face a high risk of progressing to T2DM without intervention. Therefore, early identification of high-risk populations and the implementation of personalized prevention strategies are essential. ML shows promise as a key tool in diabetes prevention. It can process large volumes of health data, such as laboratory results, lifestyle factors, and genetic information, revealing complex associations that traditional statistical methods often overlook, thereby enhancing prediction accuracy and intervention outcomes. A recent study showed that the XGBoost model accurately predicted the progression of prediabetic patients one year after intervention, using only basic clinical variables to build the model [28].

Diet is an important factor in the progression from prediabetes to T2DM. During the prediabetic stage, individuals face a significantly increased risk of developing T2DM. Although previous studies have examined the impact of various dietary patterns on this process [34], incorporating specific dietary factors into diabetes risk prediction models for personalized early intervention in clinical practice remains a complex task. Therefore, the key feature of this study is the integration of specific dietary indicators into the prediction model, which improves the accuracy of risk assessment.

During the feature selection phase, the univariate analysis and the LASSO regression model were used to select the final seven traditional indicators and nine dietary indicators for constructing models. These variables encompass metabolism, diet, body weight, age, and socioeconomic factors, providing a comprehensive assessment of the risk of progression from prediabetes to T2DM. Previous studies have shown that age, FPG, HDL-C, and BMI are crucial and robust indicators for identifying the progression [35,36]. Based on these factors, ML methods can be applied to prediction, classification, and decision-making tasks. Studies across various fields have demonstrated that ML generally outperforms traditional methods and rule-based systems in predictive performance. For example, ML models demonstrated a significantly higher AUC compared to the Lausanne Acute Stroke Registry and Analysis (ASTRAL) score in predicting acute stroke [37]. Moreover, the Bagged Trees model showed significantly faster computation times than traditional heuristic methods for predicting surgical time [38]. However, the optimal ML model often depends on data characteristics, problem complexity, and the properties of the model. In this study, the XGBoost model outperformed other models, which is similar to a recent study by Shi et al. [39].

The SHAP analysis identified FPG, eggs, whole grains, income level, red meat, nuts, HDL-C, and age as key risk factors for predicting the progression of prediabetes. Income level is rarely considered a major predictor in related studies, likely because it is not a direct biological factor causing T2DM. Instead, it influences lifestyle, environment, and healthcare access, which indirectly impact diabetes development [40]. The relationship between egg consumption and T2DM risk is complex and inconsistent, influenced by multiple factors. Our study suggests that moderate egg consumption is not significantly associated with the risk of progressing to T2DM. However, exceeding this range may increase the risk of developing the disease, consistent with the findings of Jang et al. [41,42,43,44]. Bioactive peptides in eggs, such as Egg White Hydrolysate (EWH), can activate the insulin signaling pathway and promote insulin secretion [45]. However, excessive cholesterol in egg yolk may induce inflammation, exacerbate insulin resistance, and disrupt glucose metabolism [46]. Whole grains, rich in dietary fiber, slow carbohydrate digestion and glucose absorption [47]. Heme iron in red meat promotes free radical production, leading to oxidative stress and damage to pancreatic β-cells [48]. Monounsaturated fatty acids (e.g., oleic acid in almonds) and polyunsaturated fatty acids (e.g., ω-3 fatty acids) in nuts improve cell membrane structure and function, thereby enhancing insulin receptor activity [49]. Additionally, as in previous studies, FPG, HDL-C, and age were independent predictors of progression from prediabetes to T2DM [50].

## 5. Conclusions

In this prospective cohort study, integrating dietary factors into the ML model notably enhanced the prediction of progression from prediabetes to T2DM, with the XGBoost model outperforming other models. However, this study has some limitations. First, the mean follow-up period was 3.84 years. Though this duration provides insights into short-term progression from prediabetes to T2DM, a longer-term follow-up would give a more detailed understanding of disease progression. Moreover, this study did not assess other factors that may influence the progression from prediabetes to T2DM, such as environmental factors (e.g., air pollution and life stress). Overall, this study provides valuable insights into the progression from prediabetes to T2DM in adults and highlights the importance of incorporating dietary factors. Future research should further explore long-term follow-up data, various nutritional factors, and additional potential influences to improve the precision and effectiveness of prevention and intervention strategies.

## Figures and Tables

**Figure 1 nutrients-17-00947-f001:**
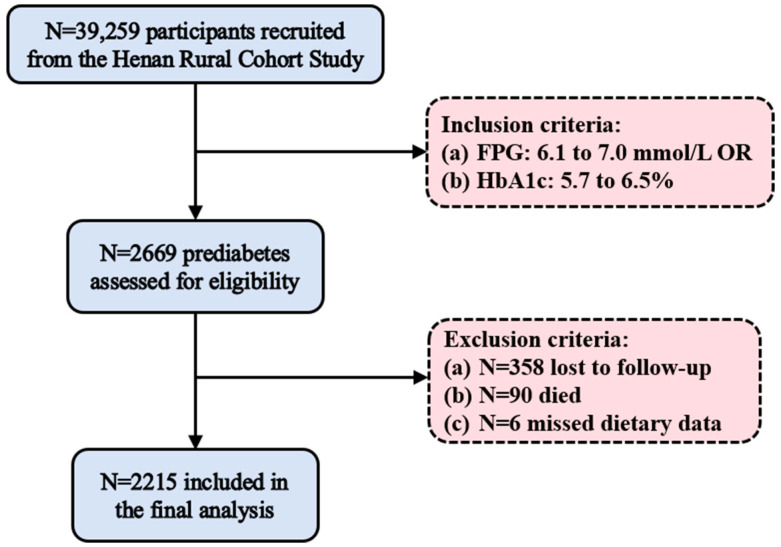
Study participant flowchart. FPG, fasting plasma glucose.

**Figure 2 nutrients-17-00947-f002:**
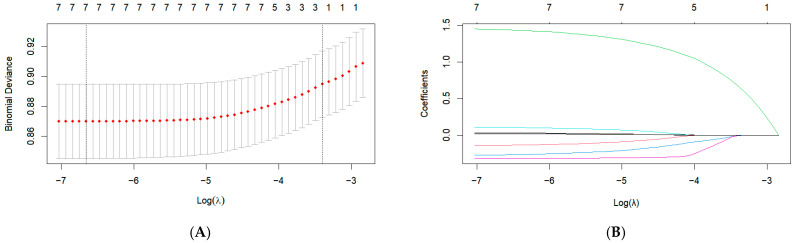
LASSO regression indicator selection. (**A**) The selection of the optimal λ parameter in the Lasso regression model through cross-validation. The left dashed line indicates the λ value that minimizes the cross-validation error, while the right dashed line represents the largest λ within one standard error of the minimum error, (**B**) The characteristics of variable coefficient variations. Each line represents the coefficient of a single variable, with each color typically corresponding to a different variable.

**Figure 3 nutrients-17-00947-f003:**
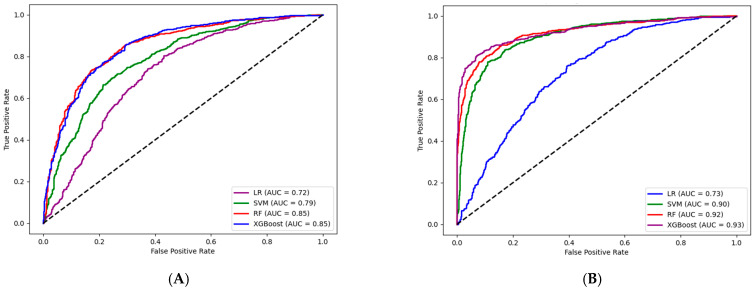
ROC curves of model based on traditional metrics (and dietary indicators). The dashed line represents a reference baseline, suggesting that the classifier’s predictions are indistinguishable from random guessing. LR, logistic regression; SVM, support vector machine; RF, random forest; XGBoost, extreme gradient boosting. (**A**) ROC curves of the model based on traditional indicators, (**B**) ROC curves of the model based on traditional indicators and dietary indicators.

**Figure 4 nutrients-17-00947-f004:**
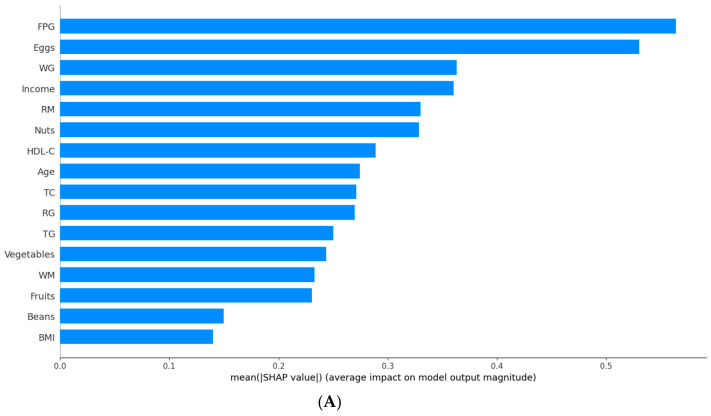
Feature importance analysis based on SHAP. FPG, fasting plasma glucose; WG, white grains; RM, red meat; HDL-C, high-density lipoprotein cholesterol; TC, total cholesterol; RG, refined grains; TG, triglycerides; WM, white meat; BMI, body mass index. (**A**) The weights of features importance, (**B**) Positive and negative impact explanation of features for predicting T2DM using SHAP values.

**Figure 5 nutrients-17-00947-f005:**
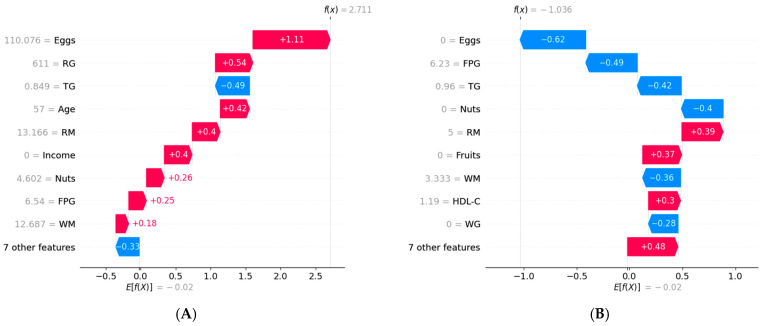
Personalized risk factor analysis based on SHAP. RG, refined grains; TG, triglycerides; RM, red meat; FPG, fasting plasma glucose; WM, white meat; HDL-C, high-density lipoprotein cholesterol; WG, white grains. (**A**) Personalized analysis of risk factors for a prediabetic patient who progressed to T2DM, (**B**) Personalized analysis of risk factors for a prediabetic patient who did not progress to T2DM.

**Table 1 nutrients-17-00947-t001:** Baseline characteristics of prediabetic (n = 2215) patients based on their glycemic status at follow-up.

Variables	Regressed to NGT (n = 1361)	Remained as Prediabetes (n = 480)	Progressed to T2DM (n = 374)	*p* Values
Age [years, M (P_25_, P_75_)]	59 (51.66)	60 (51.65)	61 (52.65)	0.096
Gender, n (%)				0.483
Female	783 (57.53%)	282 (58.75%)	228 (60.96%)	
Male	578 (42.47%)	198 (41.25%)	146 (39.04%)	
Educational levels, n (%)				**0.003**
Illiteracy and primary school	619 (45.48%)	255 (53.12%)	196 (52.41%)	
Middle school and above	742 (54.52%)	225 (46.88%)	178 (47.59%)	
Marriage, n (%)				0.921
Married/cohabiting	1221 (89.71%)	431 (89.79%)	333 (89.04%)	
Unmarried/divorced/widowed	140 (10.29%)	49 (10.21%)	41 (10.96%)	
Per capita monthly income, n (%)				0.063
<500, RMB	508 (37.32%)	181 (37.71%)	159 (42.51%)	
500~, RMB	454 (33.36%)	140 (29.17%)	122 (32.62%)	
1000~, RMB	399 (29.32%)	159 (33.12%)	93 (24.87%)	
Smoking status, n (%)				0.529
Never	968 (71.13%)	345 (71.88%)	271 (72.46%)	
Ever	136 (9.99%)	48 (10.00%)	45 (12.03%)	
Current	257 (18.88%)	87 (18.12%)	58 (15.51%)	
Drinking status, n (%)				**0.011**
Never	1007 (73.99%)	371 (77.29%)	283 (75.67%)	
Ever	51 (3.75%)	31 (6.46%)	19 (5.08%)	
Current	303 (22.26%)	78 (16.25%)	72 (19.25%)	
Physical activity, n (%)				0.182
Low	56 (4.12%)	13 (2.71%)	15 (4.01%)	
Moderate	145 (10.65%)	37 (7.71%)	33 (8.82%)	
High	1160 (85.23%)	430 (89.58%)	326 (87.17%)	
Pittsburgh Sleep Quality Index [M (P_25_, P_75_)]	3 (2,5)	3 (2,5)	3 (2,5)	0.350
Hypertension, n (%)				0.605
No	704 (51.73%)	261 (54.38%)	197 (52.67%)	
Yes	657 (48.27%)	219 (45.62%)	177 (47.33%)	
Family history of type 2 diabetes mellitus, n (%)				0.122
No	1304 (95.81%)	452 (94.17%)	350 (93.58%)	
Yes	57 (4.19%)	28 (5.83%)	24 (6.42%)	
Cancer, n (%)				0.725
No	1348 (99.04%)	475 (98.96%)	369 (98.66%)	
Yes	13 (0.96%)	5 (1.04%)	5 (1.34%)	
Kidney failure, n (%)				0.356
No	1360 (99.93%)	480 (100.00%)	373 (99.73%)	
Yes	1 (0.07%)	0 (0.00%)	1 (0.27%)	
Fasting plasma glucose [mmol/L, M (P_25_, P_75_)]	6.31 (6.20,6.56)	6.40 (6.21,6.60)	6.49 (6.30,6.70)	**<0.001**
Total cholesterol [mmol/L, M (P_25_, P_75_)]	5.10 (4.42,5.80)	4.95 (4.37,5.77)	4.83 (4.23,5.50)	**<0.001**
Triglyceride [mmol/L, M (P_25_, P_75_)]	1.49 (1.07,2.21)	1.58 (1.14,2.54)	1.74 (1.23,2.54)	**<0.001**
High-density lipoprotein cholesterol [mmol/L, M (P_25_, P_75_)]	1.26 (1.08,1.48)	1.23 (1.05,1.45)	1.19 (1.02,1.39)	**<0.001**
Low-density lipoprotein cholesterol [mmol/L, M (P_25_, P_75_)]	3.02 (2.48,3.60)	3.01 (2.51,3.69)	2.98 (2.46,3.48)	0.256
Systolic pressure [mmHg, M (P_25_, P_75_)]	132 (120,146)	130 (118,145)	132 (120,147)	0.490
Diastolic pressure [mmHg, M (P_25_, P_75_)]	82 (74,90)	80 (72,89)	81 (74,89)	0.185
Heart rate [time/minutes, M (P_25_, P_75_)]	77 (70,86)	76 (69,84)	76 (69,84)	**0.030**
Body mass index [kg/m^2^, M (P_25_, P_75_)]	25.9 (23.6,28.2)	25.9 (23.9,27.9)	26.3 (24.2,28.7)	0.107
Waist circumference [cm, M (P_25_, P_75_)]	88.1 (81.1,94.8)	87.0 (81.1,94.4)	89.0 (82.8,96.0)	0.069

NGT, normal glucose tolerance; T2DM, type 2 diabetes mellitus. Significant values are given in bold letters.

## Data Availability

Restrictions apply to the availability of these data. Data were obtained from the Henan Rural Cohort and are available from C.J.W [tjwcj2005@126.com].

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
