# Peer review of "Machine Learning Models Integrating Dietary Indicators Improve the Prediction of Progression from Prediabetes to Type 2 Diabetes Mellitus"

_nutrients, 2025, doi:10.3390/nu17060947_

Round 1

Reviewer 1 Report

Comments and Suggestions for Authors

Diabetes, and especially type 2 diabetes, is one of the diseases that are very intensively studied in a number of scientific centers around the world. Every year, many articles are published by scientists about new research on this disease. On the one hand, they present attempts to explain the causes of contracting it, on the other hand, methods of preventing its development and effects. Scientists are increasingly reaching for algorithms that could help better manage a range of data, and thus support doctors and patients in the fight against diseases.

Considering the rich literature on type 2 diabetes, the Introduction was written by the authors quite modestly. In particular, in my opinion, there is little information about the use of the titular machine model. This makes it difficult to see how the results obtained by the authors compare with previous scientific studies.

The goal that the authors set for themselves in the conducted research is also described quite generally. In my opinion, there is no clear statement as to why they chose this particular method and what effects they expected. The authors should also supplement the text with Conclusions, which should include the most important results obtained, their possible limitations and prospects for the future.

Author Response

请参阅附件。

Reviewer 2 Report

Comments and Suggestions for Authors

The authors present an analysis on predictors for prediabetes progression based on a large prospective cohort study.

The overall rationale is fine, but was mainly missed by the analysis. Title and manuscript claim, that the novel model integrates dietary factors. However, fasting glucose, HDL and income are not dietary factors.

Title: Incorrect phrasing. The novel model does not improve progression, but maybe prediction of progression.

Abstract: The methodology is too short and does not represent the actual step-by-step-approach and comparison of various models.

Introduction: Mostly fine.

For L. 60-65: The German Diabetes Risk Test and the FINDRISK score are primarily based on dietary factors. Please reference accordingly.

Methods:

Comparison of the novel models to existing models (using bio-parameters and dietary factors - such as the German DRT) is necessary. If there is a comparable Chinese index, feel free to use that one.

Results:

L. 162: The statement is imprecise or incorrect.

Tab. 1: please check correct alignment of p values (drinking status)

Please shorten decimals in accordance to plausible raw data precision (e.g. BMI and WC with one decimal).

3.2.: Why did you include more dietary variables in the model than originally showed a difference between progressors and non-progressors/remission?

Discussion: The tonality of the paper exaggerates the impact of dietary factors on the predictive power of risk scores. The differences between an index based on conventional measures and the one with dietary factors added is rather small. And: In no way is the best index based ONLY on dietary factors. Your paper wrongly suggests that on several occasions.

Comments on the Quality of English Language

rephrasing necessary on several occasions

Author Response

请参阅附件。

Round 2

Reviewer 2 Report

Comments and Suggestions for Authors

The authors have revised their manuscript in accordance to the reviewer's suggestions - with one exception.

Comparison of your novel model (including traditional markers AND dietary markers) with the performance of a traditional index (also including traditional markers AND dietary markers) would dramatically increase the interest to the readers. Currently, your readers will ask themselves, why they should use an ML tool integrating traditional markers and dietary factors. Why not use a conventional well-established index, which does that same work? Ergo, the main question actually is: does the ML approach outperform traditional multivariable indices (such as DRT or FINDRISK or any other similar index of your choice, based on classical and dietary factors).

The question, if dietary factors increase the performance when added to traditional factors in a prediction model, is very, very trivial. This has been shown repeatedly - in particular, during the development of earlier indices like FINDRISK or DRT.
